# Angular Trajectory Design for MR-SPS Using Bezier Shaping Approach

**Song Xu, Mingying Huo \*, Naiming Qi, Wenyu Feng, Tong Lin and Zheng Li**

Department of Aerospace Engineering, Harbin Institute of Technology, Harbin 150001, China
* Correspondence: huomingying@hit.edu.cn

**Abstract:** Solar power satellite (SPS) is a kind of large-scale on-orbit servicing spacecraft collecting solar energy in space and transmitting energy to the earth. The solar arrays of the SPS must point to the sun to collect enough solar energy and the antenna must point to the rectenna on the ground to transmit energy. But due to the limitation of the control effort, accurate solar and earth orientation may not be achieved. This paper focuses on the MR-SPS, and establishes the attitude kinematics and dynamics model of a MW-level MR-SPS. Angular trajectory based on Bezier shaping approach is generated at different time of a year to satisfy the control constraints. The simulation results demonstrate the effectiveness of the proposed method. Even if the control torque is limited to a small amount, the optimal angular trajectory can still ensure high average energy receiving efficiency.

**Keywords:** MR-SPS; Bezier shaping approach; angular trajectory

## 1. Introduction

With the increasing demand for energy and the depletion of non-renewable energy, the world is facing a serious energy crisis. As a kind of abundant clean and renewable energy, the rational utilization of solar energy is of great significance to solve the energy crisis and environmental problems. Solar power satellite (SPS) also called space power station or space solar power satellite (SSPS) first proposed by Glaser in 1968 [1] is a class of large-scale on-orbit servicing spacecraft that can collect solar energy in space and transmit electrical energy in the form of microwave to the earth. A typical SPS consists of three main parts: the solar collector uses the solar arrays to receive the sunlight and convert the solar energy into electric energy, some SPSs are equipped with concentrators such as lenses or concave mirror to converge the sunlight; the conversion device converts the electric energy into microwave or laser; the antenna or laser transmitter transmits the microwave or laser to the ground, in which the rectenna of the receiving station receives the microwave or laser and converts it into electric energy which can be injected to power supply system.

Over the past couple of decades, the United States, Europe, Japan, and China have put forward a variety of construction schemes for SPSs. Since Glaser first proposed the idea of building a SPS in geostationary orbit (GEO), in 1970s, NASA continued to explore the feasibility and key technologies of SPS and proposed the 1979 SPS reference system [2]. In 1990s, NASA successively proposed a series of SPS configurations, including Sun Tower, Solar Disc, Abacus and Integrated Symmetrical Concentrated system (ISC) [3]. In 2011, John Mankins came up with the solar power satellite via arbitrarily large-phased array (SPS-ALPHA) [4]. ESA put forward the Sail Tower SPS, which effectively reduced the system mass but could not achieve sustainable energy supply [5]. JAXA began the study of SPS in 1990 and several configurations were given, for example, JAXA 2001, JAXA 2002, and Tethered SPS [6], etc. Tethered SPS has smaller mass and can supply energy continuously. China has carried out research on SPS since 2015 based on two types

of SPS concept: Multirotary-joint Solar Power satellite (MR-SPS) by CAST [7] and Space Solar Power Station via Orb-shape Membrane Energy Gathering Array (SSPS-OMEGA) by Xidian University [8].

The core task of a SPS is to collect as much solar energy as possible and transmit it to the ground accurately. To accomplish the above goal, attitude of an SPS is supposed to be adjusted so that the solar collector tracks the sun and at the same time, the antenna or laser transmitter points to the rectenna in the ground accurately. Due to its large mass and inertia, large torque is required for attitude control of a SPS, which means more actuators and fuel need to be carried and further increases the mass of the system. In order to reduce fuel consumption and mass, some research has been done. [9] proposed a novel quasi-Sun-pointing attitude in Sun-frozen orbit. Although about 3% electricity must be given up, little control effort is required to deal with the solar radiation pressure and gravity-gradient torque. [10,11] developed an integrated orbit, attitude, and structural control system architecture for very large SSPS in GEO. A low-bandwidth attitude control system was proposed utilizing cyclic-disturbance accommodating control to provide precision pointing of the Abacus platform in the presence of dynamic modeling uncertainties and external disturbances. Moreover, the configuration of the actuators and the propellants required were also analyzed in detail. In [12,13], a detailed study of solar power satellites' orbit dynamics was performed, and a direct comparison between the attitude dynamics of SPS in GLP orbit and in GEO was made. When both attitude and orbit control are considered, fuel consumption is less in GLP compared to in GEO.

MR-SPS is an SPS scheme located in GEO proposed by CAST, which utilizes a number of mutually independent rotating solar arrays continuously oriented to the sun to receive solar energy and converts it into electric energy that is transmitted to the power transmission bus through a number of independent low-power conductive rotating joints. The transmission medium of the energy is microwave.

There was some research on the attitude of the MR-SPS. In [14], the flexible multi-body attitude dynamic model including external disturbances is derived, and a hybrid high/low bandwidth robust controller is proposed to achieve the attitude control of the SPS. [15,16] proposed a switched iterative learning controller (ILC) for a flexible SPS to periodically track the earth and the sun. The previous research established the dynamic model of MR-SPS based on analytical mechanics which need complicated derivatives with respect to generalized coordinates. The angular trajectory was not optimized which can cause quite large control torque.

In view of the dynamic characteristics of MR-SPS, if the microwave antenna is accurately oriented to the ground and the solar arrays are accurately oriented to the sun when the energy receiving efficiency is 100%, there will be a slew at some time around the vernal and autumnal equinox, which requires large control torques. However, subject to the total weight and degradation of the actuators, the torque generated by the actuators may fail to meet the orientation requirements. The previous attitude planning for SPS did not consider the constraints of dynamics. Aiming at this problem, an MR-SPS dynamic model was established. Using the method of Bezier shaping approach, the angular trajectory is optimized under the condition of limited control torque, which ensures the accurate earth orientation of the antenna and the quasi sun orientation of solar arrays as well as a high energy receiving efficiency. Simulation verifies the feasibility of the optimization method.

The rest of this paper is arranged as follows: In Section 2, the model of MR-SPS is presented; the angular trajectory planning based on Bezier shaping approach is described in Section 3; in Section 4, the simulation results are given. The results are summarized in Section 5.

## 2. Model Description

### 2.1. MR-SPS Structure

In terms of structure, MR-SPS can be divided into three parts, as shown in Figure 1. For a MW-level MR-SPS, there are two solar arrays in each side in the north and the south direction, each solar array is composed of 10 solar subarrays, the size of each subarray is 30 m × 100 m, and the distance between two subarrays is 2 m. The main structure is a truss structure, which consists of the upper and lower north-south main truss structure and the longitudinally connected main truss structure, with a total length of 800 m and width of 51 m. The upper north-south main truss structure is used to support the microwave transmitting antenna that is fixed to the main structure. The longitudinally connected main truss structure connects the north-south main structures together. The main structure and the antenna form the central body structure. The diameter of the microwave transmitting antenna is 140 m.

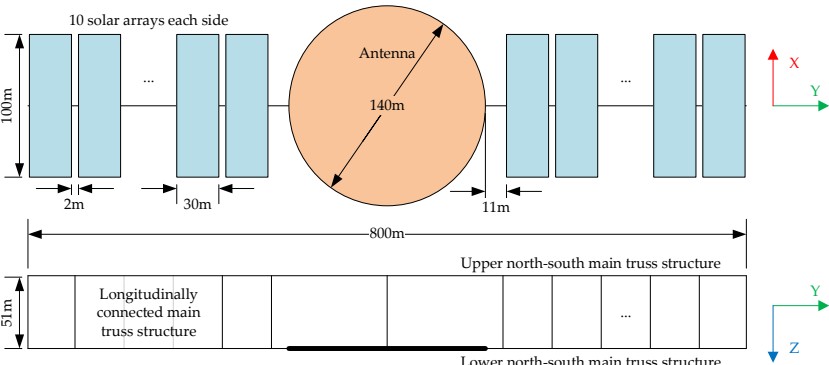

**Figure 1.** Structure of the MW-level MR-SPS.

It is assumed that the structural flexibility is not considered. Since the attitude of the 10 subarrays in each side is consistent, they can be considered to be fixed to each and form a big array. Torque actuators mounted on the central body generate three-dimensional control torque, and motors between the solar arrays and the main structure generate control torque only along the $y_a$-axis.

### 2.2. Sun and Earth Orientation Analysis

Regardless of the dynamic constraints, the real-time and precise orientation to the sun for the arrays and to the rectenna for the antenna can be achieved by adjusting the attitude of the MR-SPS and in this case, the energy receiving efficiency reaches the peak. The energy receiving efficiency is defined as the area weighted average of the cosine of the angle between the normal direction of the arrays and the solar direction.

As shown in Figure 2, five kinds of reference frame are introduced to describe the attitude of the MR-SPS.

1. The origin $o_i$ of the geocentric inertial reference frame $o_i$-$x_i y_i z_i$ is located at the center of mass of the earth, the $x_i$-axis is defined along the direction of vernal equinox, the $z_i$-axis is normal to the equatorial plane and points to the north pole, and the $y_i$-axis completes a right-handed coordinate system;

2. The origin $o_o$ of the orbital reference frame $o_o$-$x_o y_o z_o$ is at the center of mass of the main structure and antenna, the $z_o$-axis points to the center of the earth, $y_o$-axis is consistent with the negative normal direction of the orbital plane, and the $x_o$-axis completes a right-handed system;

3. The origin $o_a$ of the main structure body-fixed reference frame $o_a$ -$x_a y_a z_a$ coincides with $o_o$, the three axes are aligned with the principal axes of inertia, the $y_a$-axis is along the

longest side of the central truss, and the $z_a$-axis is perpendicular to the antenna surface.

4.  The origin $o_{cj}$ of the $j$th intermediate reference frame $o_{cj}$-$x_{cj}y_{cj}z_{cj}$ is the hinged point of the $j$th solar array and the three axes are parallel to those of the main structure frame;

5.  The origin $o_{sj}$ of the $j$th array body-fixed reference frame $o_{sj}$-$x_{sj}y_{sj}z_{sj}$ is at the center of mass of the $j$th solar array, the three axes are aligned with the principal axes of inertia, and $y_{sj}$-axis is parallel to $y_a$-axis.

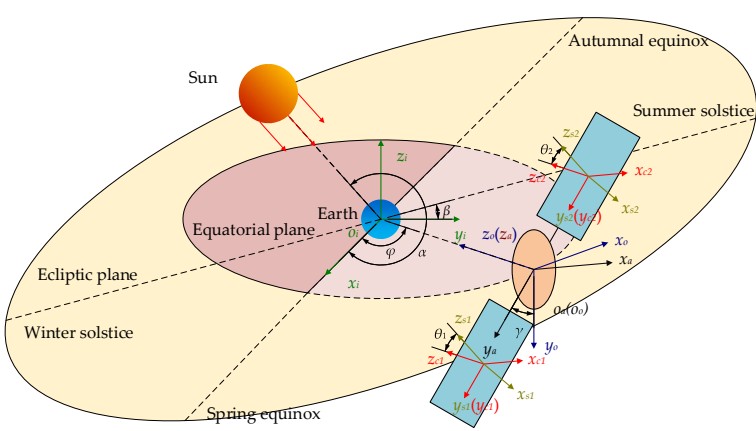

**Figure 2.** The definitions of the five reference frames.

$\varphi$ is the right ascension of the SPS, $\alpha$ is the angle that the sun runs on the ecliptic from the vernal equinox, and $\beta$ is the obliquity of the ecliptic. Limited by the area of the rectenna and the distance between the SPS and the rectenna, the antenna must point to the rectenna precisely, and in other words, the $z_a$-axis must be parallel to the $z_o$-axis at any time. Because the direction of the $z_a$-axis is determined, only one-axis motion is considered for the main structure. After rotating $\gamma$ around $z_o$-axis, $o_o$-$x_oy_oz_o$ coincides with $o_a$-$x_ay_az_a$. $\theta_j$ is the rotation angle of the $j$th array around the main truss from the initial position where $o_{sj}$-$x_{sj}y_{sj}z_{sj}$ coincides with $o_{cj}$-$x_{cj}y_{cj}z_{cj}$. $\hat{x}$ denotes the unit vector of the vector $x$ or the $x$-axis. The position unit vector of the sun relative to the earth center is $\hat{r}_s = [\cos\alpha \quad \sin\alpha\cos\beta \quad \sin\alpha\sin\beta]^T$, and the unit vector from the earth center to the centroid of the main structure and antenna is $\hat{r}_a = [\cos\varphi \quad \sin\varphi \quad 0]^T$. The two unit vectors are both expressed in the inertial frame. The $z_o$-axis is parallel to the $z_a$-axis, so $\hat{z}_a = -\hat{r}_a$. For a spacecraft in GEO, it is obvious that $\hat{y}_o = [0 \quad 0 \quad -1]^T$ and $\hat{x}_o = [-\sin\varphi \quad \cos\varphi \quad 0]^T$ expressed in the inertial frame. In fact, as long as $\gamma$ and $\theta_j$ are confirmed, the attitude of the MR-SPS is determined, and according to the geometric relationship, $\gamma$ and $\theta_j$ can be calculated by

$$\gamma = \arctan(-\frac{\hat{x}_o \cdot \hat{y}_a}{\hat{y}_o \cdot \hat{y}_a}) \tag{1}$$

$$\theta_j = \arctan(\frac{\hat{x}_a \cdot \hat{z}_{sj}}{\hat{z}_a \cdot \hat{z}_{sj}}) \tag{2}$$

For the purpose of precise solar orientation of the arrays and earth orientation of the antenna, the $y_a$-axis needs to be vertical to $\hat{r}_s$ and $\hat{r}_a$ simultaneously and the $z_{sj}$-axis

needs to be parallel to $\hat{\boldsymbol{r}}_s$ which means $\hat{\boldsymbol{z}}_{sj} = \hat{\boldsymbol{r}}_s$. In view of the symmetrical structure of the SPS, $\hat{\boldsymbol{y}}_a$ and $\hat{\boldsymbol{x}}_a$ can be obtained as follows:

$$\hat{\boldsymbol{y}}_a = \pm \frac{\hat{\boldsymbol{r}}_s \times \hat{\boldsymbol{r}}_a}{\left| \hat{\boldsymbol{r}}_s \times \hat{\boldsymbol{r}}_a \right|} \tag{3}$$

$$\hat{\boldsymbol{x}}_a = \hat{\boldsymbol{y}}_a \times \hat{\boldsymbol{z}}_a \tag{4}$$

Substitute Equations (3) and (4) into Equations (1) and (2) and in this case, the ideal angular trajectory is obtained and the energy receiving efficiency reaches to 100%.

*2.3. Attitude Dynamic Model*

The Newton–Euler approach based on the balance of all the forces and torques acting on each part of the structure is widely used to solve inverse dynamics of robots. The method avoids complicated derivatives with respect to generalized coordinates, which is common with Lagrangian-based approaches [17]. It can be adopted to model the complex dynamics of the MR-SPS.

The force balance equations of the central body and the *j*th (*j* = 1, 2) solar array are listed as follows:

$$\boldsymbol{F}_{soj} + \boldsymbol{F}_{sij} = m_{sj} \boldsymbol{C}_{sj} \boldsymbol{C}_a \boldsymbol{a}_{sj} = m_{sj} \boldsymbol{C}_{sj} \boldsymbol{C}_a (\ddot{\boldsymbol{r}}_a + \boldsymbol{C}_a^T \dot{\boldsymbol{\omega}}_a^\times \boldsymbol{r}_{cj} + \boldsymbol{C}_a^T \boldsymbol{\omega}_a^\times \boldsymbol{\omega}_a^\times \boldsymbol{r}_{cj}) \tag{5}$$

$$\boldsymbol{F}_{ao} + \sum_{k=1}^2 \boldsymbol{F}_{aik} = m_a \boldsymbol{C}_a \boldsymbol{a}_a = m_a \boldsymbol{C}_a \ddot{\boldsymbol{r}}_a \tag{6}$$

$$\boldsymbol{F}_{sij} = -\boldsymbol{C}_{sj} \boldsymbol{F}_{aij} \tag{7}$$

where $m_a$ and $m_{sj}$ are the mass of the central body and *j*th solar array respectively. $\boldsymbol{C}_{sj}$ is the rotation matrix from the central body frame to the *j*th solar array frame, and $\boldsymbol{C}_a$ denotes the orientation of the central body frame respect to the inertial frame. $\boldsymbol{F}_{ao}$ and $\boldsymbol{F}_{soj}$ are external forces acting on the central body and the *j*th solar array frame. $\boldsymbol{F}_{aij}$ and $\boldsymbol{F}_{sij}$ are the forces exerted on the central body by the *j*th solar array and on the *j*th solar array by the central body. $\boldsymbol{r}_a$ is the position vector of the central body and $\boldsymbol{r}_{sj}$ is the vector from the centroid of the central body to the centroid of the *j*th solar array expressed in the inertial frame. $\boldsymbol{\omega}_a$ denotes the angular velocity of central body respect to inertial frame expressed in the central body frame. The notation $\boldsymbol{x}^\times$ for $\boldsymbol{x} = [x_1 \quad x_2 \quad x_3]^T$ represents the skew symmetric matrix:

$$\boldsymbol{x}^\times = \begin{bmatrix} 0 & -x_3 & x_2 \\ x_3 & 0 & -x_1 \\ -x_2 & x_1 & 0 \end{bmatrix}$$

According to Equations (5)–(7), forces exerted on the central body by the *j*th solar array are given as follows:

$$\boldsymbol{F}_{aij} = \boldsymbol{A}_o^j + \boldsymbol{A}_{\dot{\omega}_a}^j \dot{\boldsymbol{\omega}}_a + \boldsymbol{A}_{other}^j \tag{8}$$

where

$$A_o^j = \frac{1}{m_a + \sum\limits_{k=1}^{2} m_{sk}} ((m_a + \sum\limits_{k=1}^{n} m_{sk}) C_{sj}^T F_{soj} - m_{sj} \sum\limits_{k=1}^{2} C_{sk}^T F_{sok} - m_{sj} F_{ao})$$

$$A_{\dot{\omega}_a}^j = \frac{m_{sj}}{m_a + \sum\limits_{k=1}^{2} m_{sk}} (-\sum\limits_{k=1}^{2} (m_{sk} r_{ck}) + (m_a + \sum\limits_{k=1}^{n} m_{sk}) r_{cj})^\times$$

$$A_{other}^j = \frac{m_{sj} \omega_a^\times \omega_a^\times}{m_a + \sum\limits_{k=1}^{2} m_{sk}} (\sum\limits_{k=1}^{2} (m_{sk} r_{ck}) - (m_a + \sum\limits_{k=1}^{n} m_{sk}) r_{cj})$$

The external forces acting on the MR-SPS include gravity, solar pressure, and microwave radiation force. The magnitude of solar pressure and microwave radiation pressure is far less than that of gravity, so they can be ignored. The external can be expressed as follows:

$$F_{ao} = -\frac{\mu m_a}{|r_a|^3} C_a r_a \tag{9}$$

$$F_{soj} = -\frac{\mu m_{sj}}{|r_a + C_a^T r_{ci}|^3} (C_{sj} C_a r_a + C_{sj} r_{cj}) \tag{10}$$

where $\mu$ denotes the gravitational coefficient of the earth.

The torque balance equations of the central body and the $j$th ($j$ = 1,2) solar array are listed as follows:

$$T_c + T_{ao} + \sum\nolimits_{k=1}^{2} (T_{aik} + r_{ck}^\times F_{aik}) = I_a \dot{\omega}_a + \omega_a^\times I_a \omega_a \tag{11}$$

$$T_{soj} + T_{sij} = I_{sj}(-\omega_{sj}^\times C_{sj} \omega_a + C_{sj} \dot{\omega}_a + \dot{\omega}_{sj}) + (C_{sj} \omega_a + \omega_{sj})^\times I_{sj}(C_{sj} \omega_a + \omega_{sj}) \tag{12}$$

$$T_{sij} = -C_{sj} T_{aij} \tag{13}$$

$$\tau_{sij} = D_j^T T_{sij} \tag{14}$$

where $I_a = \mathrm{diag}(I_{a1}, I_{a2}, I_{a3})$ denotes the moment of inertia of the central body relative to the center of mass, and $I_{sj} = \mathrm{diag}(I_{sjx}, I_{sjy}, I_{sjz})$ denotes the moment of inertia of the $j$th solar array relative to the center of mass. $T_c$ and $\tau_{scj}$ are the control torques of the central body and the $j$th solar array. $T_{aoj}$ and $T_{soj}$ are external disturbance torques acting on the central body and the $j$th solar array. $T_{aij}$ and $T_{sij}$ are the torques exerted on the central body by the $j$th solar array and on the $j$th solar array by the central body. $\omega_{sj} = D_j \dot{\theta}_j$ denotes the angular velocity of the $j$th solar array respect to the central body frame expressed in $j$th solar array frame, where $D_j = [0 \quad 1 \quad 0]^T$ represents the unit vector of the direction of the rotation axis of the $j$th solar array and $\theta_j$ represents the angle that the $j$th solar array rotates. $C_{sj}$ can be written as:

$$C_{sj} = \begin{bmatrix} \cos\theta_j & 0 & -\sin\theta_j \\ 0 & 1 & 0 \\ \sin\theta_j & 0 & \cos\theta_j \end{bmatrix} \tag{15}$$

The attitude dynamics are expressed in a form convenient for the subsequent research on attitude control. Substituting Equations (8), (13), and (14) into Equations (11) and (12), the attitude dynamics can be expressed as follows:

$$\boldsymbol{B}^0_{\dot{\boldsymbol{\omega}}_a}\dot{\boldsymbol{\omega}}_a + \boldsymbol{B}^0_{\ddot{\theta}_1}\ddot{\theta}_1 + \boldsymbol{B}^0_{\ddot{\theta}_2}\ddot{\theta}_2 + \boldsymbol{B}^0_{other} = \boldsymbol{T}_c + \boldsymbol{B}^0_o \tag{16}$$

$$\boldsymbol{B}^j_{\dot{\boldsymbol{\omega}}_a}\dot{\boldsymbol{\omega}}_a + \boldsymbol{B}^j_{\ddot{\theta}_j}\ddot{\theta}_j + \boldsymbol{B}^j_{other} = \tau_{scj} + \boldsymbol{B}^j_o \tag{17}$$

where

$$\boldsymbol{B}^0_o = \boldsymbol{T}_{ao} + \sum_{k=1}^{n}(\boldsymbol{r}^{\times}_{ck}\boldsymbol{A}^k_o + \boldsymbol{C}^T_{sk}\boldsymbol{T}_{sok}) \quad \boldsymbol{B}^j_o = \boldsymbol{D}^T_j\boldsymbol{T}_{soj}$$

$$\boldsymbol{B}^0_{\dot{\boldsymbol{\omega}}_a} = \boldsymbol{I}_a + \sum_{k=1}^{n}(\boldsymbol{C}^T_{sk}\boldsymbol{I}_{sk}\boldsymbol{C}_{sk} - \boldsymbol{r}^{\times}_{ck}\boldsymbol{A}^k_{\dot{\boldsymbol{\omega}}_a}) \quad \boldsymbol{B}^j_{\dot{\boldsymbol{\omega}}_a} = \boldsymbol{D}^T_j\boldsymbol{I}_{sj}\boldsymbol{C}_{sj}$$

$$\boldsymbol{B}^0_{\ddot{\theta}_j} = \boldsymbol{C}^T_{sj}\boldsymbol{I}_{sj}\boldsymbol{D}_j \quad \boldsymbol{B}^j_{\ddot{\theta}_j} = \boldsymbol{D}^T_j\boldsymbol{C}_{sj}\boldsymbol{D}_j$$

$$\boldsymbol{B}^0_{other} = \boldsymbol{\omega}^{\times}_a\boldsymbol{I}_a\boldsymbol{\omega}_a + \sum_{k=1}^{n}\boldsymbol{C}^T_{sk}(-\boldsymbol{I}_{sk}\boldsymbol{\omega}^{\times}_{sk}\boldsymbol{C}_{sk}\boldsymbol{\omega}_a + (\boldsymbol{C}_{sk}\boldsymbol{\omega}_a + \boldsymbol{\omega}_{sk})^{\times}\boldsymbol{I}_{sk}(\boldsymbol{C}_{sk}\boldsymbol{\omega}_a + \boldsymbol{\omega}_{sk}) - \boldsymbol{r}^{\times}_{ck}\boldsymbol{A}^k_{other})$$

$$\boldsymbol{B}^j_{other} = \boldsymbol{D}^T_j\boldsymbol{C}_{sj}(-\boldsymbol{I}_{sj}\boldsymbol{\omega}^{\times}_{sj}\boldsymbol{C}_{sj}\boldsymbol{\omega}_a + (\boldsymbol{C}_{sj}\boldsymbol{\omega}_a + \boldsymbol{\omega}_{sj})^{\times}\boldsymbol{I}_{sj}(\boldsymbol{C}_{sj}\boldsymbol{\omega}_a + \boldsymbol{\omega}_{sj}))$$

The external torques acting on the MR-SPS include gravity gradient torque, solar pressure torque, and microwave radiation torque caused by the deviation of centroid and pressure center. Due to the symmetrical structure, the deviation is small, so the magnitude of solar pressure torque and microwave radiation torque is far less than that of gravity gradient torque and cannot be known in advance, so they can be ignored. The external torques can be expressed as follows:

$$\boldsymbol{T}_{ao} = \frac{3\mu}{|\boldsymbol{r}_a|^5}(\boldsymbol{C}_a\boldsymbol{r}_a)^{\times}\boldsymbol{I}_a(\boldsymbol{C}_a\boldsymbol{r}_a) \tag{18}$$

$$\boldsymbol{T}_{soj} = \frac{3\mu}{|\boldsymbol{C}_a\boldsymbol{r}_a + \boldsymbol{r}_{cj}|^5}(\boldsymbol{C}_{sj}\boldsymbol{C}_a\boldsymbol{r}_a + \boldsymbol{C}_{sj}\boldsymbol{r}_{cj})^{\times}\boldsymbol{I}_{sj}(\boldsymbol{C}_{sj}\boldsymbol{C}_a\boldsymbol{r}_a + \boldsymbol{C}_{sj}\boldsymbol{r}_{cj}) \tag{19}$$

## 3. Trajectory Planning for the MR-SPS Considering Dynamic Constraints

### 3.1. State Approximation Using Bezier Curve

In Bezier shaping approach, it is assumed in advance that the angular trajectory of the MR-SPS follows Bezier curve functions. The approximation of $\gamma$ and $\theta_j$ can be expanded by applying Bezier curve functions. To avoid repetition, only the processing approach for $\gamma$ is given, and $\theta_j$ can be processed in the same way. The approximation of $\gamma$ is expanded as follows:

$$\gamma(\tau) = \sum_{k=0}^{n_\gamma}B_{\gamma,k}(\tau)P_{\gamma,k} \tag{20}$$

where $\tau = t/T(0 \le \tau \le 1)$ is the scaled time, and $T$ is the flight time; $n_\gamma$ is the order of the Bezier curve function. $P_{\gamma,k}(k \in [0, n_\gamma])$ are the Bezier coefficients; $B_{\gamma,k}(\tau)$ are the Bezier basis functions given by:

$$B_{\gamma,k}(\tau) = \frac{n_\gamma!}{k!(n_\gamma - k)!}\tau^k(1-\tau)^{n_\gamma - k} \tag{21}$$

The first and second $\tau$-derivative of $\gamma$ can be derived as follows:

$$\gamma'(\tau) = \sum_{k=0}^{n_\gamma} B'_{\gamma,k}(\tau)P_{\gamma,k} \tag{22}$$

$$\gamma''(\tau) = \sum_{k=0}^{n_\gamma} B''_{\gamma,k}(\tau)P_{\gamma,k} \tag{23}$$

where $B'_{\gamma,k}(\tau)$ and $B''_{\gamma,k}(\tau)$ are first and second $\tau$-derivative of the Bezier basis functions that can be derived according to Equation (20) as:

$$B'_{\gamma,k}(\tau) = \begin{cases} -n_\gamma(1-\tau)^{n_\gamma-1} & k=0 \\[2mm] \dfrac{n_\gamma!}{(k-1)!(n_\gamma-k)!}\tau^{k-1}(1-\tau)^{n_\gamma-k} & \\[1mm] \quad -\dfrac{n_\gamma!}{k!(n_\gamma-k-1)!}\tau^{k}(1-\tau)^{n_\gamma-k-1} & k\in[1,n_\gamma-1] \\[2mm] n_\gamma\tau^{n_\gamma-1} & k=n_r \end{cases} \tag{24}$$

$$B''_{\gamma,k}(\tau) = \begin{cases} n_\gamma(n_\gamma-1)(1-\tau)^{n_\gamma-2} & k=0 \\[1mm] n_\gamma(n_\gamma-1)(n_\gamma-2)\tau(1-\tau)^{n_\gamma-3} - 2n_\gamma(n_\gamma-1)(1-\tau)^{n_\gamma-2} & k=1 \\[1mm] \dfrac{n_\gamma!}{(k-2)!(n_\gamma-k)!}\tau^{k-2}(1-\tau)^{n_\gamma-k} - \dfrac{2n_\gamma!}{(k-1)!(n_\gamma-k-1)!}\tau^{k-1}(1-\tau)^{n_\gamma-k-1} & \\[1mm] \quad +\dfrac{n_r!}{k!(n_\gamma-k-2)!}\tau^{k}(1-\tau)^{n_\gamma-k-2} & k\in[2,n_\gamma-2] \\[1mm] n_\gamma(n_\gamma-1)(n_\gamma-2)\tau^{n_\gamma-3}(1-\tau) - 2n_\gamma(n_\gamma-1)\tau^{n_\gamma-2} & k=n_\gamma-1 \\[1mm] n_\gamma(n_\gamma-1)\tau^{n_\gamma-2} & k=n_\gamma \end{cases} \tag{25}$$

The start time is selected as when the right ascension between the sun and the SPS is about 90° because the value of $\gamma$ is equal to the inclination of the sun at equatorial plane and $\dot\gamma$ is equal to 0. After a solar day (86400s) instead of the orbital period considering the movement of the sun, $\gamma$ and $\dot\gamma$ are almost equal to the ones at start time. So, $T$ is chosen as a solar day. To ensure the convergence, the following boundary conditions are required to be satisfied:

$$\begin{aligned} \gamma(\tau=0) &= \gamma_0 & \gamma(\tau=1) &= \gamma_f \\ \gamma'(\tau=0) &= T\dot\gamma_0 & \gamma'(\tau=1) &= T\dot\gamma_f \end{aligned} \tag{26}$$

where $\dot{x}$ denotes the derivative of $x$ with respect to time $t$; $x'$ denotes the derivative of $x$ with respect to scaled time $\tau$. $\gamma_0$, $\gamma_f$, $\dot\gamma_0$, and $\dot\gamma_f$ can be calculated according to Equations (1)–(4) so that the approximation of $\gamma$ $\dot\gamma$ is continuous after a solar day.

Substituting $\tau=0$ and $\tau=1$ into Equations (20)–(22) and (26), four Bezier coefficients $P_{\gamma,0}$, $P_{\gamma,1}$, $P_{\gamma,n_\gamma-1}$, and $P_{\gamma,n_\gamma}$ can be determined as follows:

$$\begin{aligned} P_{\gamma,0} &= \gamma_0 \\ P_{\gamma,1} &= \gamma_0 + T\dot\gamma_0/n_\gamma \\ P_{\gamma,n_\gamma} &= \gamma_f - T\dot\gamma_f/n_\gamma \\ P_{\gamma,n_\gamma} &= \gamma_f \end{aligned} \tag{27}$$

In order to meet the dynamic constraints at every moment, some discrete points can be adopted to calculate the dynamic constraints. The discrete points can be obtained by applying Gaussian Legendre distribution, which is defined as the roots of *m*-order Legendre polynomial to be scaled to $[0,1]$ and to be arranged in ascending order as follows: $[\tau]_{m \times 1} = [\tau_1 \quad \tau_2 \quad \cdots \quad \tau_{m-1} \quad \tau_m]^T$. The discrete points of $\gamma$ and its first and second $\tau$-derivatives can be given in compact matrix notation form as follows:

$$[\gamma]_{m \times 1} = [B_\gamma]_{m \times (n_\gamma + 1)} [P_\gamma]_{(n_\gamma + 1) \times 1} \tag{28}$$

$$[\gamma']_{m \times 1} = [B'_\gamma]_{m \times (n_\gamma + 1)} [P_\gamma]_{(n_\gamma + 1) \times 1} \tag{29}$$

$$[\gamma'']_{m \times 1} = [B''_\gamma]_{m \times (n_\gamma + 1)} [P_\gamma]_{(n_\gamma + 1) \times 1} \tag{30}$$

where $[P_\gamma]_{(n_\gamma + 1) \times 1} = [P_{\gamma,0} \quad P_{\gamma,1} \quad [X_\gamma]^T_{(n_\gamma - 3) \times 1} \quad P_{\gamma,n_\gamma - 1} \quad P_{\gamma,n_\gamma}]^T$ is the Bezier coefficient column matrix and $[X_\gamma] = [P_{\gamma,2} \quad \cdots \quad P_{\gamma,n_r - 1}]^T$ is the unknown coefficient column matrix to be optimized. $[B_\gamma]_{m \times (n_\gamma + 1)}$, $[B'_\gamma]_{m \times (n_\gamma + 1)}$, and $[B''_\gamma]_{m \times (n_\gamma + 1)}$ are Bezier basis function matrix and the first and second $\tau$-derivative matrices of the Bezier basis function which can be calculated by substituting $[\tau]_{m \times 1}$ into Equations (21), (24), and (25).

Once the order of Bezier curve and the number of discrete points are determined, $[B_\gamma]_{m \times (n_\gamma + 1)}$, $[B'_\gamma]_{m \times (n_\gamma + 1)}$, and $[B''_\gamma]_{m \times (n_\gamma + 1)}$ are determined, only calculated once and stored. In the process of optimization, they can be used in every iteration without repeated calculation, which reduce the calculation burden of the computer.

### 3.2. Description of the Optimization Problem

As a large-scale spacecraft, the MR-SPS requires large torque in the attitude tracking process. During a short period of time around the time of the equinox, if the arrays strictly point to the sun at any moment, there will be a large angle of ration in a short time for the main structure when the right ascension between the sun and the SPS is about 0° or 180°, resulting in large control effort. From the perspective of economic benefits, the large-scale spacecraft needs to limit the mass. The larger the maximum control torque is, the heavier the actuators are which leads to an increase in the mass of actuators as well as the launch cost. Moreover, supposing that the actuators fail, the maximum output torque decreased, which causes degradation of the output of the actuator. Therefore, there is a need to plan the angular trajectory when the control torque is limited.

From the geometric relationship shown in Figure 2, after rotating $-\pi/2$ around *x*-axis and $-\varphi-\pi/2$ around *y*-axis, the inertial frame is parallel to the orbital frame. The orbit frame is parallel to the main structure frame after it rotates $\gamma$ around *z*-axis. The *j*th intermediate frame is parallel to the *j*th array frame after rotating $\theta_j$ around *y*-axis. The rotation matrix from the inertial frame to the *j*th array frame can be written as:

$$C_j = C_{sj} C_a = C_y(\theta_j) C_z(\gamma) C_y(-\varphi - \pi/2) C_x(-\pi/2) =$$
$$\begin{bmatrix} \cos\varphi\sin\theta_j - \sin\varphi\cos\gamma\cos\theta_j & \sin\varphi\sin\theta_j + \cos\varphi\cos\gamma\cos\theta_j & -\sin\gamma\cos\theta_j \\ \sin\varphi\sin\gamma & -\cos\varphi\sin\gamma & -\cos\gamma \\ -\cos\varphi\cos\theta_j - \sin\varphi\cos\gamma\sin\theta_j & -\sin\varphi\cos\theta_j + \cos\varphi\cos\gamma\sin\theta_j & -\sin\gamma\sin\theta_j \end{bmatrix} \tag{31}$$

One rotation can be expressed by a unit quaternion $Q = [q_0 \quad q^T]^T$, where $q_0 \in \mathbf{R}$ and $q \in \mathbf{R}^3$ are the scalar and vector part of the unit quaternion respectively. The rotation of the main structure frame relative to the inertial frame can be expressed as:

$$\boldsymbol{Q}_a = [q_{a0} \quad \boldsymbol{q}_a^T]^T = \begin{bmatrix} \sqrt{2}/2\cos(\varphi/2+\gamma/2+\pi/4) \\ -\sqrt{2}/2\cos(\varphi/2-\gamma/2+\pi/4) \\ -\sqrt{2}/2\cos(\varphi/2-\gamma/2-\pi/4) \\ \sqrt{2}/2\cos(\varphi/2+\gamma/2-\pi/4) \end{bmatrix} \tag{32}$$

The first and second derivatives can be expressed as:

$$\dot{\boldsymbol{Q}}_a = \begin{bmatrix} -\sqrt{2}/4\sin(\varphi/2+\gamma/2+\pi/4)(\dot{\varphi}+\dot{\gamma}) \\ \sqrt{2}/4\sin(\varphi/2-\gamma/2+\pi/4)(\dot{\varphi}-\dot{\gamma}) \\ \sqrt{2}/4\sin(\varphi/2-\gamma/2-\pi/4)(\dot{\varphi}-\dot{\gamma}) \\ -\sqrt{2}/4\sin(\varphi/2+\gamma/2-\pi/4)(\dot{\varphi}+\dot{\gamma}) \end{bmatrix} \tag{33}$$

$$\ddot{\boldsymbol{Q}}_a = \begin{bmatrix} -\sqrt{2}/4\sin(\varphi/2+\gamma/2+\pi/4)(\ddot{\varphi}+\ddot{\gamma})-\sqrt{2}/8\cos(\varphi/2+\gamma/2+\pi/4)(\dot{\varphi}+\dot{\gamma})^2 \\ \sqrt{2}/4\sin(\varphi/2-\gamma/2+\pi/4)(\ddot{\varphi}-\ddot{\gamma})+\sqrt{2}/8\cos(\varphi/2-\gamma/2+\pi/4)(\dot{\varphi}-\dot{\gamma})^2 \\ \sqrt{2}/4\sin(\varphi/2-\gamma/2-\pi/4)(\ddot{\varphi}-\ddot{\gamma})+\sqrt{2}/8\cos(\varphi/2-\gamma/2-\pi/4)(\dot{\varphi}-\dot{\gamma})^2 \\ -\sqrt{2}/4\sin(\varphi/2+\gamma/2-\pi/4)(\ddot{\varphi}+\ddot{\gamma})-\sqrt{2}/8\cos(\varphi/2+\gamma/2-\pi/4)(\dot{\varphi}+\dot{\gamma})^2 \end{bmatrix} \tag{34}$$

The attitude kinematics equation of the main structure can be expressed as follows:

$$\dot{\boldsymbol{Q}}_a = \frac{1}{2}\boldsymbol{G}_a\boldsymbol{\omega}_a \tag{35}$$

where $\boldsymbol{G}_a = [-\boldsymbol{q}_a \quad (q_{a0}\boldsymbol{E}_3 + \boldsymbol{q}_a^\times)^T]^T$, and it is easy to find that $\boldsymbol{G}_a^T\boldsymbol{G}_a = \boldsymbol{E}_3$ where $E_3$ is the identity matrix. So $\boldsymbol{\omega}_a$ and $\dot{\boldsymbol{\omega}}_a$ can be expressed as:

$$\boldsymbol{\omega}_a = 2\boldsymbol{G}_a^T\dot{\boldsymbol{Q}}_a \tag{36}$$

$$\dot{\boldsymbol{\omega}}_a = 2\dot{\boldsymbol{G}}_a^T\dot{\boldsymbol{Q}}_a + 2\boldsymbol{G}_a^T\ddot{\boldsymbol{Q}}_a \tag{37}$$

Substituting Equations (28)–(34), (36), and (37) into Equations (16) and (17), the compact matrix notation form of $T_c$ and $\tau_{scj}$ can be written as $[\boldsymbol{T}_c]_{3m\times1}$ and $[\tau_{scj}]_{m\times1}$. Compared with $T_c$, the value of $\tau_{scj}$ is much smaller, so only the value of $T_c$ is constrained. Considering that the maximum torque of $T_c$ is $\boldsymbol{T}_{c\max} = [T_{c1\max} \quad T_{c2\max} \quad T_{c3\max}]^T$ and the minimum torque of $T_c$ is $\boldsymbol{T}_{c\min} = [T_{c1\min} \quad T_{c2\min} \quad T_{c3\min}]^T$, 6 $m$ inequality constraints are obtained as follows:

$$\boldsymbol{1}_{m\times1} \otimes \boldsymbol{T}_{c\min} \leq [\boldsymbol{T}_c]_{3m\times1} \leq \boldsymbol{1}_{m\times1} \otimes \boldsymbol{T}_{c\max} \tag{38}$$

where $\boldsymbol{1}_{m\times1}$ is a matrix with all the elements equal to 1. $\otimes$ denotes the Kronecker product operator.

In theory, if the integrated attitude energy control system [18] is employed for the MR-SPS, the process of generating the control torque does not consume any energy. Continuous and efficient energy supply is the main task, so the energy receiving efficiency is selected as the performance index. The cosine of the angle between the normal vector of the array and the vector of the sun can measure the energy receiving efficiency. The following equation can calculate the average energy receiving efficiency in a solar day for one array:

$$\eta_j = \int_0^T \hat{\boldsymbol{z}}_{sj}^T \boldsymbol{C}_j \hat{\boldsymbol{r}}_s dt = \int_0^1 \hat{\boldsymbol{z}}_{sj}^T \boldsymbol{C}_j \hat{\boldsymbol{r}}_s d\tau \tag{39}$$

By calculation at the discretization points, the continuous angular trajectory optimization problem for the MR-SPS can be converted into the following small-scale nonlinear programming problem:

$$
\begin{aligned}
\min \quad & J = \sum_{k=1}^{2} \eta_k \\
& [X_\gamma]_{(n_\gamma-3)\times 1} \quad [X_{\theta_j}]_{(n_\theta-3)\times 1}(j=1,2) \\
\text{s.t.} \quad & \mathbf{1}_{m\times 1} \otimes \mathbf{T}_{c\min} \leq [\mathbf{T}_c]_{3m\times 1} \leq \mathbf{1}_{m\times 1} \otimes \mathbf{T}_{c\max}
\end{aligned}
\tag{40}
$$

$[X_\gamma]_{(n_\gamma-3)\times 1}$ and $[X_{\theta_j}]_{(n_\theta-3)\times 1}(j=1,2)$ are the unknown Bezier coefficients, a total of $n_r + 2n_\theta - 9$ variables need to be optimized.

*3.3. Initial Value Selection*

The selection of the initial value of the variables has a great impact on the result of the optimization. The initial value close to the optimal solution can avoid local optimization and improve the operation speed. Section 2.2 has given the ideal angular trajectory, namely the analytical angular trajectory without dynamic constraints which can be used as a reference for the selection of initial value. At most time, by calculating ideal $\gamma$ and $\theta_j$ at discrete time, the compact matrix notation forms $[\gamma_{initial}]_{m\times 1}$ and $[\theta_{initialj}]_{m\times 1}$ are obtained. The initial values $[X_{\gamma initial}]_{(n_\gamma-3)\times 1}$ and $[X_{\theta_j initial}]_{(n_\theta-3)\times 1}(j=1,2)$ can be calculated by solving the following equation to get the least square solution:

$$
[\gamma_{initial}]_{m\times 1} = [B_\gamma]_{m\times(n_\gamma+1)}[P_{\gamma,0} \quad P_{\gamma,1} \quad [X_{\gamma initial}]_{(n_\gamma-3)\times 1}^{T} \quad P_{\gamma,n_\gamma-1} \quad P_{\gamma,n_\gamma}]^{T}
\tag{41}
$$

$$
[\theta_{initialj}]_{m\times 1} = [B_\theta]_{m\times(n_\theta+1)}[P_{\theta_j,0} \quad P_{\theta_j,1} \quad [X_{\theta_j initial}]_{(n_\theta-3)\times 1}^{T} \quad P_{\theta_j,n_\theta-1} \quad P_{\theta_j,n_\theta}]^{T}
\tag{42}
$$

However, around equinox, the rate of change of the ideal angular trajectory is large when the right ascension between the sun and the SPS is about 0° or 180°, so the inversely calculated initial value violates the dynamic constraints a lot. Nevertheless, at this moment, the altitude angle of the sun is small, and quite high energy receiving efficiency is guaranteed as long as the $y_a$-axis is perpendicular to the orbital plane. $[\gamma_{initial}]_{m\times 1}$ and $[\theta_{initialj}]_{m\times 1}$ are given by: $\gamma = 0$ and $\theta_j = -\pi/2 + 2\pi\tau$. By solving Equations (41) and (42), the initial value $[X_{\gamma initial}]_{(n_\gamma-3)\times 1}$ and $[X_{\theta_j initial}]_{(n_\theta-3)\times 1}(j=1,2)$ can be acquired.

By solving the optimization using two sets of initial value, the better result is selected as the optimal solution. The following criterion can help choose the initial value:

$$
\sigma = \int_0^1 \ddot{\gamma}^2 d\tau
\tag{43}
$$

where $\sigma$ represents the smoothness of the curve.

## 4. Simulation Results

To verify the effectiveness of the proposed Bezier curve-based shaping approach, the trajectory planning for the MR-SPS in different time of the year is conducted. The order of the Bezier curve function is selected as $n_\gamma = n_\theta = 20$ and the number of LG discrete points is selected as $m = 50$. The nonlinear programming problem is solved using sequential quadratic programming. The parameters of the MR-SPS are listed in Table 1.

**Table 1.** Parameters of the MR-SPS.

| Parameters | Value |
| --- | --- |
| $I_a$ | diag $(4.4 \times 10^9,\ 1 \times 10^8,\ 4.4 \times 10^9) \mathrm{kg \cdot m}^2$ |
| $I_{sj}$ | diag $(1.88 \times 10^8,\ 2.08 \times 10^7,\ 2.08 \times 10^8) \mathrm{kg \cdot m}^2$ |
| $m_a$ | $8 \times 10^4 \mathrm{kg}$ |
| $m_{sj}$ | $2.5 \times 10^4 \mathrm{kg}$ |
| $\boldsymbol{r}_{c1}$ | $[0 \quad 230 \quad -30]^T \mathrm{m}$ |
| $\boldsymbol{r}_{c2}$ | $[0 \quad -230 \quad -30]^T \mathrm{m}$ |

*4.1. Angular Trajectory around Solstice*

Assume that the MR-SPS is on orbit at 18 June 2025 and the maximum torque and minimum torque are set as $\boldsymbol{T}_{c\max} = [80 \quad 80 \quad 80]^T \mathrm{N \cdot m}$ and $\boldsymbol{T}_{c\min} = [-80 \quad -80 \quad -80]^T \mathrm{N \cdot m}$. The optimal angular trajectory of $\gamma$ and $\theta_j$ generated by the proposed Bezier approach and the ideal angular trajectory of $\gamma$ and $\theta_j$ are shown in Figure 3. It can be obtained that the optimal angular trajectory generated by the Bezier approach is very close to one ideal trajectory because the rate of change of the rotation angle is always small and the control effort is enough to track the ideal trajectory. The average energy receiving efficiency is 99.97% which is very close to 100%.

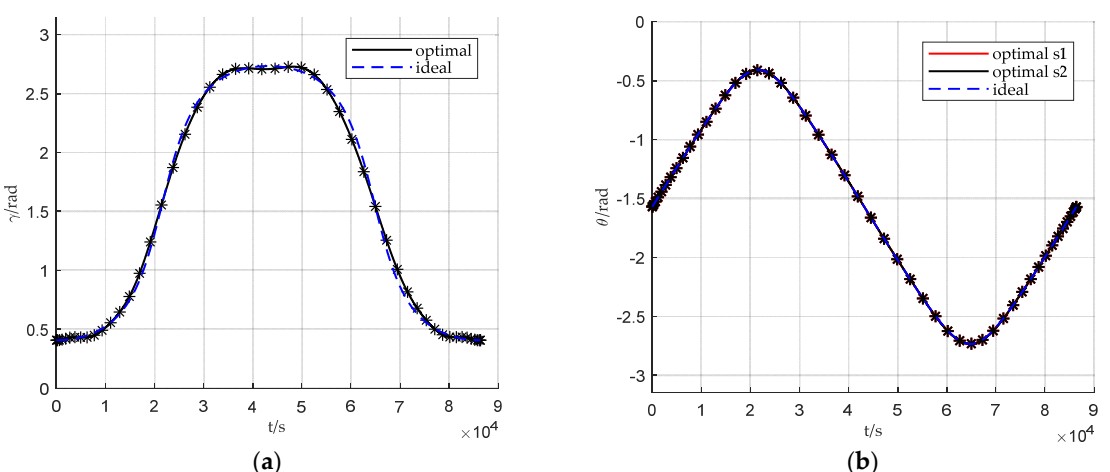

(**a**)  (**b**)

**Figure 3.** Angular trajectory around solstice. (**a**) Angular trajectory of $\gamma$; (**b**) angular trajectory of $\theta_j$.

The control torque history is shown in Figure 4. It can be observed that all the components of $\boldsymbol{T}_c$ satisfy the constraints, and the peak value of $\boldsymbol{T}_c$ of the ideal trajectory is 150 N·m and is larger than that of the optimal trajectory which is 80 N·m. Although the peak value of control torque is less for the optimal trajectory, the loss of the efficiency is negligible. The value of the component $T_{cy}$ is much less than the other components. The histories of $\tau_{sc1}$ and $\tau_{sc2}$ are almost the same due to the symmetrical structure and the history of $\tau_{scj}$ of the optimal trajectory and ideal trajectory is similar. If the constraints are less strict, the ideal trajectory can also be applied.

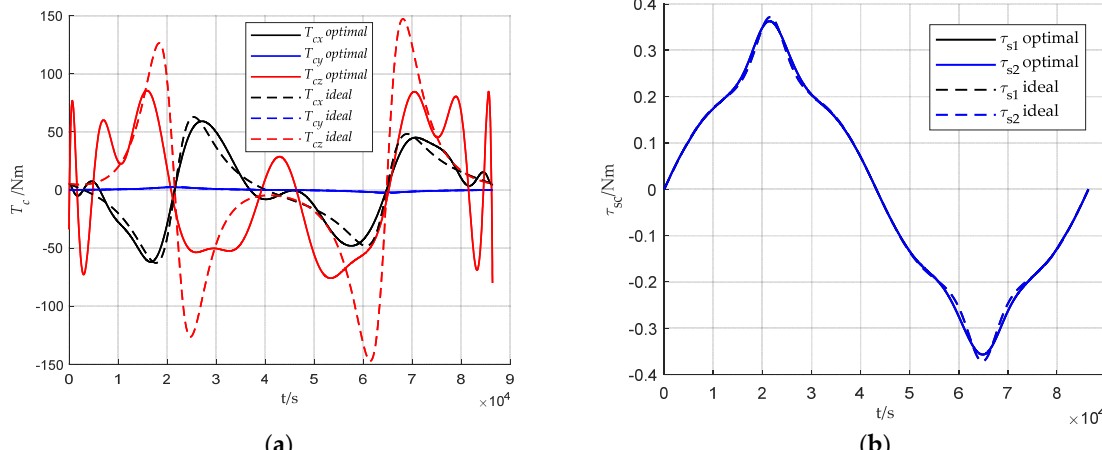

**Figure 4.** Control torque history around solstice. (**a**) Control torque history of the components of $T_c$; (**b**) Control torque history of $\tau_{scj}$.

### 4.2. Angular Trajectory around Equinox

Assume that the MR-SPS is on orbit at 15 September 2025 and the maximum control torque and minimum control torque are set as $T_{c\max} = [100 \quad 100 \quad 100]^T \, \text{N} \cdot \text{m}$ and $T_{c\min} = [-100 \quad -100 \quad -100]^T \, \text{N} \cdot \text{m}$. The optimal angular trajectory of $\gamma$ and $\theta_j$ generated by the proposed Bezier approach and the ideal angular trajectory are shown in Figure 5. It can be obtained that the optimal angular trajectory generated by the Bezier approach is very close to one ideal trajectory at the beginning and the end but in the middle it switches to another ideal trajectory, because the rate of change of the rotation angle is quite large at about 21,600 s and 64,800 s and the control effort is not enough to track the ideal trajectory. The average energy receiving efficiency is 99.97% which is very close to 100%.

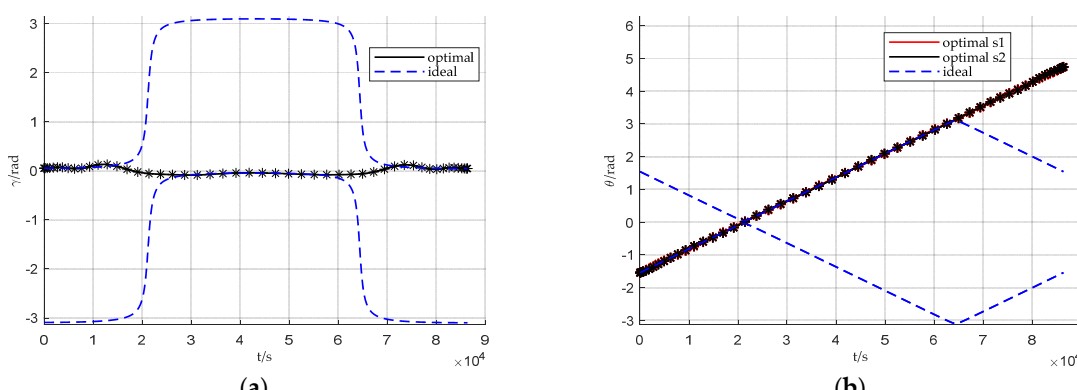

**Figure 5.** Angular trajectory around equinox. (**a**) Angular trajectory of $\gamma$; (**b**) angular trajectory of $\theta_j$.

The control torque history is shown in Figure 6. It can be observed that all the components of $T_c$ satisfy the constraints of the optimal trajectory, and the peak value of $T_c$ of the ideal trajectory is over $10^4$ N·m and is far larger than that of the optimal trajectory which is 100 N·m. The value of the component $T_{cy}$ is much less than the other components. The history of $\tau_{sc1}$ and $\tau_{sc2}$ is almost the same due to the symmetrical structure but the history of $\tau_{scj}$ of the optimal trajectory and ideal trajectory is different during the period when the trajectory is switched.

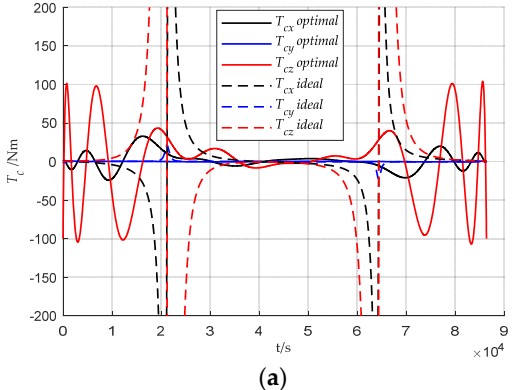
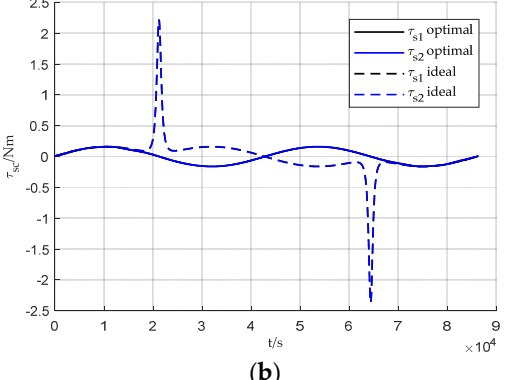

<div align="center">(<strong>a</strong>)</div>

<div align="center">(<strong>b</strong>)</div>

**Figure 6.** Control torque history around equinox. (**a**) Control torque history of the components of $T_c$; (**b**) control torque history of $\tau_{scj}$.

## 5. Conclusions

The generation of angular trajectory for a MW-level MR-SPS using Bezier shaping approach is demonstrated in this paper. The precise solar and earth orientation for the MR-SPS is analyzed and the ideal angular trajectory without dynamic constraints is given which is helpful for the initial value selection of the following optimization. The attitude dynamics and kinematics are modeled for calculating the inequality constraints of the optimization. The angular trajectory of the MR-SPS is assumed to be described by Bezier curve functions. The angular trajectory design around equinox and solstice is performed. The numerical simulation verifies that the Bezier shaping approach can generate an angular trajectory that meet the dynamic constraints. The energy receiving efficiency can be near 100% and the demand for torque actuators is reduced.

**Author Contributions:** Conceptualization, S.X.; methodology, S.X. and M.H.; software, S.X. and M.H.; validation, S.X. and W.F.; formal analysis, S.X. and M.H.; investigation, S.X. and W.F.; resources, S.X.; data curation, T.L.; writing—original draft preparation, S.X.; writing—review and editing, Z.L. and T.L.; visualization Z.L. and S.X.; supervision, N.Q. and M.H.; project administration, N.Q.; funding acquisition, N.Q. All authors have read and agreed to the published version of the manuscript.

**Funding:** This research was funded in part by the National Science Foundation of China, grant number 11702072; in part by the China Postdoctoral Science Foundation, grant number 2017M611372; in part by the Heilongjiang Postdoctoral Fund, grant number LBH-Z16082; and in part by the National Natural Science Foundation of China, grant number 61903245.

**Institutional Review Board Statement:** Not applicable.

**Informed Consent Statement:** Not applicable.

**Data Availability Statement:** Not applicable.

**Acknowledgments:** Not applicable.

**Conflicts of Interest:** The authors declare no conflict of interest.

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
