# Peer review of "Angular Trajectory Design for MR-SPS Using Bezier Shaping Approach"

_aerospace, doi:10.3390/aerospace9100528_

Round 1

Reviewer 1 Report

The paper studies the problem of the angular motion construction for the SPS. The idealized model without solar panels flexible effect is considered. The one-axis satellite rotation as well as relative solar panel rotation are based on the Bezier curves. The are a number of remarks:

1. The first and second reference frames introduction is not correct. In general, inertial frame is based on the vernal equinox and equatorial plane for some epoch (e.g. J2000). The frame introduction in paper is based on the orbit plane which position can't be defined before the inertial frame is introduced. The orbital frame is based on the radius-vector and velocity which is also incorrect since in case non-circular orbit (which is idealization even for the case of keplerian motion) these two vectors are not orthogonal. It is better to use the radius vector and orbit plane.

2. What is the right ascension of the GEO satellite? Maybe the latitude is meant?

3. It is not clear why the flexibility effects are not considered as well as other perturbations in the numerical modelling section. 

4. It is also not clear what is 100% of the efficiency. The normal to solar panel lies in the orbit plane, so there is non-zero angle between the normal and Sun direction, so the power output is not on the maximum.

5. Why only the one-axis satellite motion is considered? It seems that the better performance will be when normal to the solar panel is directed to the Sun.

Reviewer 2 Report

See the attachment.

Round 2

Reviewer 1 Report

Authors answered to all remarks. Paper can be accepted.